# Depression, Anxiety and Stress in Health Professionals in the COVID-19 Context

**DOI:** 10.3390/ijerph19074402

**Published:** 2022-04-06

**Authors:** Gracielle Pereira Aires Garcia, Isabela Fernanda Larios Fracarolli, Heloisa Ehmke Cardoso dos Santos, Samuel Andrade de Oliveira, Bianca Gonzalez Martins, Lacir José Santin Junior, Maria Helena Palucci Marziale, Fernanda Ludmilla Rossi Rocha

**Affiliations:** 1School of Nursing of Ribeirão Preto, University of São Paulo (EERP/USP), São Paulo 14040-902, Brazil; isabela_larios@hotmail.com (I.F.L.F.); heloisa.santos@usp.br (H.E.C.d.S.); samuelandrade@usp.br (S.A.d.O.); lacirsantin@usp.br (L.J.S.J.); marziale@eerp.usp.br (M.H.P.M.); ferocha@eerp.usp.br (F.L.R.R.); 2School of Pharmaceutical Sciences, São Paulo State University (UNESP), São Paulo 01049-010, Brazil; biagmarts@gmail.com

**Keywords:** occupational health, health personnel, depression, anxiety, psychological stress, COVID-19

## Abstract

To assess the prevalence of depression, anxiety and stress symptoms in health professionals in the COVID-19 pandemic context. Method: Cross-sectional study with non-probabilistic (snow-ball) sampling method. The assessment was performed using the Depression, Anxiety, and Stress Scale (DASS-21) and the prevalence of symptoms severity was calculated by point and 95% confidence interval. The analysis of the psychometric properties of DASS-21 was performed using confirmatory factor analysis (CFA) and the following goodness of fit indices: χ^2^/df (chi-square ratio by degrees of freedom), Tucker–Lewis index (TLI), comparative fit index (CFI) and root mean square error of approximation (RMSEA) with a 90% confidence interval. Results: The study participants were 529 health professionals (82.4% women and 66.7% nursing professionals). CFA of the DASS-21 structural model presented adequate fit for the sample (χ^2^/df = 3.530; CFI = 0.979; TLI = 0.976; RMSEA = 0.069). Regarding prevalence, moderate to extremely severe symptoms of depression, anxiety and stress were found in 48.6%, 55.0% and 47.9% of the participants, respectively. Conclusion: The use of DASS-21 confirmed the validity and reliability of the data. The prevalence of depression, anxiety and stress symptoms in the participants indicated a high risk of mental illness in health professionals in the COVID-19 pandemic.

## 1. Introduction

The COVID-19 pandemic worsened the adversities related to the health professionals’ work context, contributing to the deterioration of their physical and mental health, especially as a result of work overload and of the stressors faced daily [1], such as long working hours, low pay, lack of professional recognition and high risk of contamination [2]. Among health professionals, the nursing category stands out as being especially vulnerable to psychological distress due to the constant exposure to the physical and psychological demands of patients contaminated by the novel coronavirus [3,4].

In this context, challenges for the health services include the following: the exponential increase in demands, leading to the risk of saturation of the health systems; the need to reorganize health units and hospital sectors; inadequate provision and scarcity of Personal Protective Equipment (PPE) for the safety and protection of professionals; inadequate personnel management in view of the high quantity of sick leave of professionals or those belonging to risk groups for COVID-19; the need to learn about new disease prevention and treatment protocols and the rapid changes presented by new scientific evidence; the assistance provided to critically ill patients; and the individual confrontation of this situation, which has caused fear of contagion and virus spread among health professionals in view of the increase in the number of infection cases [1,5,6,7].

Scientific evidence has revealed that health professionals are prone to developing suffering and psychological disorders such as stress, anxiety, depression and burnout syndromes due to the characteristics of their daily work [2,8], mainly during the COVID-19 pandemic [9,10,11,12,13]. In addition, high levels of exhaustion, irritability and insomnia, reduced empathy, a decline in cognitive functions and work performance, decreased appetite or indigestion, nervousness, frequent crying, and suicidal thoughts have been observed [3,14,15].

Symptoms of depression are characterized by reduced self-esteem, apathy and anhedonia, while anxiety is characterized by acceleration and anticipation of future events; stress is associated with situations in which symptoms of excitement and/or tension persist as a result of the individual’s inability to use coping strategies [16,17]. It is also recognized that anxiety and stress are natural and biological phenomena that can be more severe when the time and intensity of their manifestations exceed the physiological limit of the individual [18].

Despite having different theoretical definitions, depression and anxiety can share similar and non-specific symptoms and are related in a complex way, at different points of the same continuum, which can favor the evolution from one condition to another over time. As with anxiety and depression, there are nonspecific symptoms that make it difficult to determine whether anxiety or stress disorders are present [16].

Therefore, this study was carried out with the objective of evaluating the prevalence of stress, anxiety and depression symptoms in health professionals in the COVID-19 pandemic context.

## 2. Materials and Methods

### 2.1. Study Design and Sample

This was a cross-sectional and observational study, with a non-probabilistic (snow-ball) sampling method. To calculate the minimum sample size, the need for 5 to 10 subjects per evaluated parameter of the instrument was considered [19], in addition to a dropout rate of approximately 20%. Thus, considering the 45 parameters of DASS-21 (21 items, 21 errors and three correlations between factors), the minimum required sample size was from 282 to 563 participants.

### 2.2. Procedures and Data Collection

The data were collected from August to October 2020. The health professionals were invited to participate in the study by announcing the research in social networks of national reach, such as Facebook and WhatsApp. The invitation message contained a link to access the data collection instruments available on the Google Forms electronic platform. The professionals who agreed to participate in the study signed the Informed Consent Form (ICF). Inclusion criteria were: (a) health professionals including physicians; nurses, nurse technicians and assistants; and physiotherapists; (b) physicians, nurses, nurse technicians/assistants and physiotherapists working in COVID-19 clinics. This study was approved by the Research Ethics Committee (CAAE 34650620.3.0000.5393). It followed ethical regulations established by Resolution 466/2012 of the Brazilian National Health Council [20].

### 2.3. Instruments

To characterize the participants, the tool used was an instrument consisting of individual factors (age, gender, marital status, number of children), occupational factors (profession, experience time, weekly workload, double employment), workers’ health conditions and issues related to the COVID-19 pandemic.

In relation to the assessment of depression, anxiety and stress symptoms, the reduced version of the Depression, Anxiety and Stress Scale (DASS-21) developed by Lovibond and Lovibond (1995) [17] was used. DASS-21 has 21 items and three dimensions: Depression (items 3, 5, 10, 13, 16, 17, 21); Anxiety (items 2, 4, 7, 9, 15, 19, 20); and Stress (items 1, 6, 8, 11, 12, 14, 18); and has a four-point Likert-type answer scale, varying from 0 (did not apply at all) to 3 (applied a lot or most of the times) [17]. In this research, the Portuguese version of DASS-21 proposed by Vignola and Tucci (2014) [21] was used.

### 2.4. Data Analysis

The psychometric properties of DASS-21 applied to the sample were analyzed by estimating the factorial, convergent and discriminant construct validity, in addition to the factorial invariance and the reliability of the data. Before performing these analyses, the psychometric sensitivity of the instrument’s items was confirmed using form measures (skewness and kurtosis) of items’ responses. Absolute skewness values lower than three and kurtosis lower than seven were indicative of non-violation of the normality assumption, which indicates adequate psychometric sensitivity of the evaluated items [22].

Factorial validity was tested using confirmatory factor analysis (CFA) with the robust weighted least squares estimation method adjusted for mean and variance (WLSMV). The following goodness of fit indices of the structural model used were: χ^2^/df (ratio of chi-square by degrees of freedom), Tucker–Lewis index (TLI), comparative fit index (CFI) and root mean square error of approximation (RMSEA) with a 90% confidence interval [22,23], considered adequate if χ^2^/df ≤ 2.0 [24]; GFI and CFI ≥ 0.90 [25]; and RMSEA ≤ 0.10 [22,23]. Factor weights (λ) were considered adequate when ≥ 0.50 [19]. Modification indices were calculated using the Lagrange multipliers (LM) method, considering values of LM > 11 (*p* < 0.001).

Factorial invariance between independent samples was evaluated using multigroup cross-validation analysis and the CFI difference statistical test (ΔCFI). For this purpose, the sample was randomly divided into two independent samples (test *n* = 271; validation *n* = 258). The CFI values of the configurational, metric and scalar models were considered, respectively, with models whose CFI reduction was less than 0.01 being considered invariant [26].

To analyze the convergent validity, the average variance extracted (AVE) of each DASS-21 factor was estimated, and considered adequate if AVE ≥ 0.50 [19,27]. Discriminant validity was accepted when the AVE for each factor was larger than the squared Pearson correlation between the two factors (AVEi and AVEj ≥ rij2) [27]. 

The data reliability was estimated using the composite reliability (CR) and the alpha ordinal coefficient (α), considering CR and α values ≥ 0.70 as indicators of acceptable reliability [28].

For the statistical analyses, the IBM SPSS Statistics 22 (IBM Corp., Armonk, NY, USA) and MPLUS 7.2 (Muthén and Muthén, Los Angeles, CA, USA) programs were used. 

Prevalence of the severity of the depression, anxiety and stress symptoms was calculated by point and 95% confidence interval, following the recommendations of the scale authors. Thus, initially, the answers given by each participant for each factor of the instrument were added up; subsequently, the scores obtained were multiplied by two and, finally, the severity classification of each individual was determined, as follows: Depression (>27 = extremely severe depression; 27–21 = severe depression; 20–14 = moderate depression; 13–10 = mild depression; 9–0 = no depression/normal); Anxiety (>19 = extremely severe anxiety; 19–15 = severe anxiety; 14–10 = moderate anxiety; 9–8 = mild anxiety; 7–0 = no anxiety/normal); Stress (>33 = extremely severe stress; 33–26 = severe stress; 25–19 = moderate stress; 18–15 = mild stress; 14–0 = no stress/normal) [17]. 

## 3. Results

### 3.1. Characteristics of Participants

The characteristics of the 529 participants are summarized in Table 1. The results show the predominance of female professionals (82.4%) and nurses (66.7%); the main health care units were first aid/emergency care (26.0%) and adult or pediatric wards (17.0%); most of the participants (67.1%) reported having only one employment and daytime work shifts (50.2%).

Regarding working conditions, it was found that the mean workload of the participants was 44.8 working hours per week; 380 (71.8%) participants considered that the institution’s physical structure was not adequate for healthcare during the pandemic; 419 (79.2%) indicated that the number of professionals was insufficient to meet the demand of the health services; 238 (45.0%) stated that the institution does not provide clear communication about the healthcare of suspected or confirmed patients with COVID-19; 378 (71.5%) reported that the leaders were not prepared to work with the team; 340 (64.2%) stated that the institution did not offer professional training for the care of patients with COVID-19 and 267 (50.5%) indicated that PPE was offered in insufficient number and quality.

Regarding factors related to the physical and mental health, the participants considered their health status to be very good (85, 16.1%), good (238, 45.0%) and moderate (161, 30.4%); 95 (18.0%) professionals reported having been diagnosed with depression and 78 (15.0%) with another type of psychological disorder; 91 (17.2%) reported having respiratory problems; 76 (14.3%) obesity; 73 (14.0%) musculoskeletal disorders; 47 (9.0%) arterial hypertension; 17 (3.2%) heart problems and 16 (3.0%) reported diabetes mellitus. It was also observed that 402 (75.9%) participants were not diagnosed with COVID-19; however, 40.0% of the professionals were removed from work due to suspicion of the disease. A total of 402 (75.9%) participants reported fear of being infected by SARS-CoV-2; 506 (95.6%) participants reported that they had at least one family member belonging to the risk group for COVID-19; 209 (39.5%) participants reported feeling unappreciated at work; 360 (68.0%) answered that there was no type of psychological support offered to the workers by the institution in the pandemic context.

### 3.2. Psychometric Characteristics of the Measuring Instrument 

The items of DASS-21 presented adequate psychometric sensitivity (skewness = −0.13–1.31; kurtosis = −1.25–0.98). CFA of the DASS-21 structural model presented excellent fit to the sample (χ^2^/df = 3.530; CFI = 0.979; TLI = 0.976; RMSEA = 0.069; CI 90% = [0.063–0.075]), with factor weights (λ) ≥ 0.645 and strong correlations between the Depression, Anxiety and Stress factors (r ≥ 0.865). Figure 1 presents the CFA of the refined model to the sample.

In relation to the convergent validity of the factors, it proved to be adequate [AVE(Depression) = 0.705; AVE(Anxiety) = 0.603; AVE(Stress) = 0.651]. However, no discriminant validity was observed between the factors (r^2^ ≥ 0.748), which is justified by the high correlation between them. Adequate reliability of the data was also verified (CR ≥ 0.913 and α# ≥ 0.911). Factorial invariance between independent samples was attested (ΔCFImetric-configurational = 0.001; ΔCFIscalar-metric = 0.001), confirming the stability of the proposed model. The CFA of the DASS-21 to the different samples is shown in Table 2.

### 3.3. Prevalence of Depression, Anxiety and Stress Symptoms in the Sample

A high prevalence of symptoms was evidenced in the sample (Table 3), highlighting the moderate to extreme severity of depression (48.6%), anxiety (55.0%) and stress (47.9%) symptoms among the health professionals.

## 4. Discussion

The aim of the present study was to evaluate the prevalence of stress, anxiety and depression symptoms in Brazilian health professionals in the COVID-19 pandemic context. Of the 529 health professionals participating in the study, 66.7% were nursing workers. It should be noted that nursing is the largest occupational group in the health workforce. It is composed of nurses, nursing technicians and nursing assistants (with higher, technical and basic educational levels, respectively), accounting for approximately 70% of the health professionals (17% nurses, 53% nursing technicians and assistants), followed by physicians (15.70%), dentists (9%), pharmacists (4.9%) and midwives (0.2%) in Brazil [29,30]). In the pandemic, nursing professionals are the main frontline healthcare providers working against COVID-19. 

Epidemic studies have shown that health professionals are at increased risk for psychological and mental disorders, due to the worsening of working conditions caused by the COVID-19 pandemic [15,31,32]. This study presents the high prevalence of depression, anxiety and stress symptoms in health professionals, reinforcing the negative psychological impact of the pandemic on mental health. In addition, the excellent psychometric properties of DASS-21 applied to the sample were shown. The results corroborate current scientific evidence.

A study that investigated depression, anxiety and stress symptoms among physicians living in Ethiopia showed prevalence values of 37.7%, 39.0% and 44.2%, respectively [33]. In Oman, 32.3% of the health professionals who participated in a cross-sectional study reported having depressive symptoms, 34.1% reported anxiety and 23.8% suffered from stress [34]. 

Of a total of 1244 physicians and nurses from five general hospitals in Belgium, a study identified moderate to extremely severe symptoms of depression in 28.8% of the participants, anxiety in 41.8% and stress in 25.1% of the professionals [35], with nurses and female participants reporting anxiety symptoms more frequently than physicians and male participants, respectively (that is, 63.2% of the nurses had symptoms vs. 23.5% of the physicians, *p* < 0.001; and 57.4% of the female participants had symptoms vs. 33.6% of the male participants, *p* < 0.001).

In order to assess the psychological impact of the COVID-19 pandemic among health professionals, a study carried out in intensive care units of hospitals in Ireland found positive scores in 201 workers (42.6%) for depression and in 213 (45.1%) for anxiety and stress [36]. Among 208 Nepali health care workers, 62 (30%) participants were positive for anxiety, 47 (22.5%) for depression and 25 (12%) for stress; a higher prevalence of depression 18 (30%) and stress 10 (17%) was found in nurses compared to paramedics [37].

In the USA, a study examined the prevalence of emotional distress among nurses working in South Dakota during the COVID-19 pandemic and verified that general emotional distress was reported by 22.2% of the participants, while anxiety, depression and stress were symptoms reported by 15.8%, 14.5% and 11.9% of the nurses, respectively [38].

A systematic review of anxiety in health professionals identified an overall prevalence of 35%, which was higher in women and among nursing professionals when compared to physicians [39]. A Portuguese study carried out with nurses also revealed higher prevalence of depression, anxiety and stress symptoms among women [40]. Thus, the nursing professionals stand out as a risk group for the occurrence of work-related mental disorders.

Among the main risk factors related to worse mental health outcomes, a study carried out among physicians and nurses in Belgium identified the following: being a nurse or a young professional, remaining isolated and having an increased workload; in addition, even higher levels of burnout, insomnia and anxiety were found among nurses when compared to physicians [35]. A research study carried out among physicians in Turkey also found that women and less experienced persons were more vulnerable to the occurrence of mental health problems [41]. The same study also showed that front line physicians who experienced increased weekly working hours and increased demand from COVID-19 patients, less support from colleagues and supervisors, little organizational support, and lower feelings of competence presented a higher risk of mental health impairment. 

Another study on the worsening of anxiety symptoms in health professionals in the COVID-19 context observed a significant influence of the pandemic on the development and aggravation of psychological disorders, with the most affected workers being those who work on the front line of patient care, especially nursing technicians, nurses and physicians [42], in addition to those who already have some chronic disease [39].

In addition to nursing professionals, a study conducted in Brazil during the pandemic revealed a high prevalence of depression, anxiety and stress symptoms among other health professionals such as dentists, pharmacists, nutritionists and psychologists [43]. Furthermore, another study confirmed symptoms of moderate to severe anxiety in multi-professional residents, with a significant association with work in COVID-19 care units and direct healthcare for suspected/confirmed cases of the disease, and younger residents, who were already undergoing psychological counseling after entering the residency and who were in continuous use of psychotropic drugs [44]. 

The severity of depression, anxiety and stress symptoms observed in the health professionals of the present study reveal how challenging the pandemic context has been, contributing to the mental illness of these individuals [45]. The high prevalence of depression, anxiety and stress, simultaneously, can be explained by the interrelation and overlapping of these psychological symptoms, and is not always specific to the determination of each of these disorders [16].

In addition, from the temporal orientation theory [46,47], it is recognized that manifestations of depression, anxiety and stress refer, respectively, to projections involving the past, the future and the present. Regarding health professionals, the COVID-19 pandemic determined greater mental suffering and negative feelings related to all these events, such as isolation; fear of contamination by the SARS-CoV-2 virus and death; possibility of contamination and loss of family, friends and co-workers; insecurity related to the lack of knowledge and professional training about COVID-19 treatment and prevention protocols, especially at the beginning of the pandemic; professional depreciation; and uncertainties related to an indeterminate future [31,40,48,49]. Added to these feelings are the precarious working conditions of health professionals and the daily confrontation of extremely stressful situations in the work environment caused by the pandemic, such as overload and long working hours, shortage of personal protective equipment, decrease in the number of workers, and inadequate physical structure of health services to meet the demand and complexity of patients [35,50], which were conditions reported by the participants of this study.

It is worth mentioning that the psychological impacts caused by pandemics were synthesized in a literature review study that evidenced an increase in suicide cases, greater aggression among people and a greater occurrence of cases of acute stress in epidemics prior to the one caused by COVID-19 [51]. Therefore, the importance of a healthy psychosocial work environment to enhance health professionals’ job satisfaction and to avoid psychological distress during the COVID-19 pandemic is highlighted [52].

In relation to the psychometric properties of DASS-21 applied to the health professionals in this study, CFA showed adequate validity and reliability of the data. Thus, the factorial validity of DASS-21 for the sample was confirmed, with the refined model consisting of three factors and 21 items, which presented adequate factorial weights and strong correlations between the Depression, Anxiety and Stress factors. This model presented excellent fit indices for the sample, a result that corroborates findings in other validation studies of the same instrument for different contexts [43,53,54,55,56]. The high correlation between the Depression, Anxiety and Stress factors may explain the lack of discriminant validity of DASS-21 applied to the sample [53].

Reinforcing the results of this study, a systematic review carried out in 2019 evaluated the psychometric properties of DASS-21 used in 48 studies and found that the instrument has robust psychometric properties and wide applicability, and that it can be used to identify depression, anxiety and stress symptoms in different contexts and populations [57]. On the other hand, a validation study of DASS-21 among 1532 health professionals from three hospitals in China showed that the latent structure of the instrument in the sample was better represented by a unifactorial model [58].

It should be clarified that this study has limitations, such as its cross-sectional study design, which does not allow inferring cause and effect relationships. However, from the application of DASS-21 among health professionals, it was possible to know the reality experienced by these individuals in the challenging COVID-19 pandemic context. In addition, the validity and reliability of the data obtained was confirmed, which proves the accuracy of the results. It is also noteworthy that the diverse information obtained may serve as a guide for managers and authorities to develop intervention and mental health care strategies aimed at health professionals in the context of global health crises.

## 5. Conclusions

The prevalence of depression, anxiety and stress symptoms among the participants indicated a high risk of mental illness of health professionals in the COVID-19 pandemic context. In this sense, the results raise an alarm about the need to adopt strategies to mitigate risk factors that trigger mental suffering and psychological disorders in health professionals; these strategies must be immediately implemented in order to promote the mental health of these individuals.

In addition, it should be noted that safety at work represents a universal right of every worker today, in line with Goal 8—Decent work and economic growth of the United Nations’ 2030 Agenda for Sustainable Development, which is a global commitment undertaken by 193 countries, including Brazil, and proposes the action of governments, institutions, companies and society in general to face the greatest challenges of contemporaneity.

## Figures and Tables

**Figure 1 ijerph-19-04402-f001:**
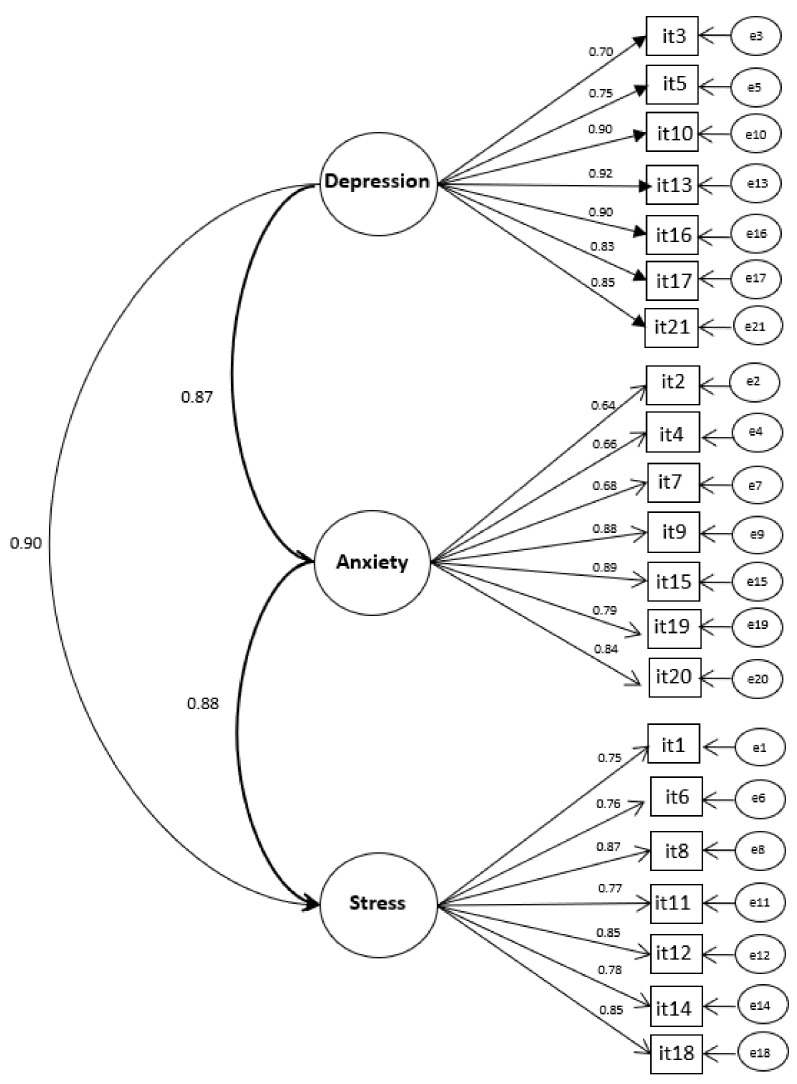
Factorial model of the Depression, Anxiety and Stress Scale for the sample.

**Table 1 ijerph-19-04402-t001:** Individual and occupational status of participants (*n* = 529).

Characteristics	*n (%)*
Gender	
Female	436 (82.5)
Male	93 (17.5)
Professional Category	
Nurse	353 (66.7)
Nurse technicians	92 (17.3)
Nurse assistants	02 (0.3)
Physician	59 (11.1)
Physiotherapist	23 (4.3)
Health care unit #	
First Aid/Emergency	138 (26.0)
Adult or pediatric ward	112 (21.1)
Adult, pediatric or neonatal Intensive Care Unit (ICU)	98 (18.5)
Exclusive ICU for COVID-19 care	58 (10.9)
Exclusive ward for COVID-19 care	90 (17.0)
Surgical center and material and sterilization center	54 (10.2)
Oncology units	25 (4.7)
Administrative sectors	47 (8.8)
Employment Contract(s)	
More than one job	174 (32.9)
Only one job	355 (67.1)
Work shift #	
Morning	139 (26.2)
Afternoon	107 (20.2)
Night	140 (26.4)
Daytime (morning and afternoon)	266 (50.2)
Evening	85 (16.0)

Note: ^#^ more than one response possibility.

**Table 2 ijerph-19-04402-t002:** Confirmatory factor analysis and reliability of the DASS-21 to different samples.

		Confirmatory Factor Analysis								Reliability		
Sample	*n*	λ	χ^2^	df	χ^2^/df	RMSEA	90% CI	CFI	TLI	AVE (D/A/S)	CR (D/A/S)	α# (D/A/S)
Total sample	529	0.645–0.923	656.583	186	3.530	0.069	0.063–0.075	0.979	0.976	0.705/0.603/0.651	0.943/0.913/0.929	0.94/0.91/0.92
DASS test	271	0.588–0.915	427.824	186	2.300	0.069	0.061–0.078	0.976	0.973	0.706/0.566/0.636	0.944/0.900/0.924	0.94/0.90/0.92
DASS validation	258	0.626–0.941	363.971	186	1.957	0.061	0.052–0.070	0.986	0.984	0.710/0.644 0.669	0.944/0.926/0.934	0.94/0.92/0.93

λ: Factor weights; χ^2^/df: Ratio of chi-square to degrees of freedom; RMSEA: root mean square error of approximation; 90% CI: 90% confidence interval; CFI: comparative fit index; TLI: Tukey–Lewis Index; AVE: average variance extracted; CR: composite reliability; α#: alpha ordinal coefficient; D/A/S: Depression, Anxiety and Stress factors; DASS-21: Depression, Anxiety and Stress Scale.

**Table 3 ijerph-19-04402-t003:** Prevalence (point and 95% confidence interval) of the depression, anxiety and stress symptoms in the sample.

Severity	Depression		Anxiety		Stress	
	*n*	%[95% CI]	*n*	%[95% CI]	*n*	%[95% CI]
Normal	201	38.0 [33.9–42.1]	196	37.0 [33.0–41.2]	204	38.6 [34.4–42.8]
Mild	71	13.4 [10.5–16.3]	42	7.9 [5.6–10.2]	72	13.6 [10.7–16.5]
Moderate	100	18.9 [15.6–22.2]	112	21.2 [17.7–24.7]	92	17.4 [14.2–20.6]
Severe	71	13.4 [10.5–16.3]	43	8.1 [5.8–10.4]	104	19.7 [16.3–23.1]
Extremely severe	86	16.3 [13.1–19.5]	136	25.7 [22.0–29.4]	57	10.8 [8.2–13.4]
Total	529	100	529	100	529	100

Note: 95% CI: 95% confidence interval.

## Data Availability

The data can be found on the SurveyMonkey platform.

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
