# Peer review of "Depression, Anxiety and Stress in Health Professionals in the COVID-19 Context"

_ijerph, 2022, doi:10.3390/ijerph19074402_

Round 1
Reviewer 1 Report
Congratulation to the authors for the nice work conducted. I really enjoy reading this document. The document is well structured and consistent with the object of research.
Introduction: the author/s provide data on the importance of expansion in this field of research.
Methodology: this section explains how the study was carried out and details the research design and measures used.
Results: this section explains the results obtained in an orderly and concise form, being easy to understand and consistent with what was stated in the theoretical framework.
Discussion: An analysis of the results is made and it is related to other studies so that the importance of the data obtained can be seen.
The author/s have made a good job, although I contribute here some suggestions for the improvement of the quality of the document:
- In the Introduction part: grammar check by a native speaker.
- In the Discussion part: add more previous studies to confirm or not the results obtained.
- Explain the possible practical applications of the study carried out.
Author Response
Dear Reviewer,
We would like to thank you all for your attention during the review process of our manuscript. We are very grateful for all the corrections and suggestions made, which contributed greatly to its improvement. We hope to have carried out all the adjustments requested by the reviewers and met their expectations. We remain at our disposal for any necessary clarification.
AUTHORS’ ANSWERS
- We apologize for the mistakes initially made by the translators in the Portuguese-English version. In accordance with the requests of all reviewers, the manuscript was submitted to an extensive English revision (see the attached translation certificate).
- A scarcity of scientific evidence on the assessment of the prevalence of anxiety, depression and stress among health professionals using the DASS-21 was identified. But two more studies were added in the Discussion. Thank you.
- We believe that the confirmation of the validity of the DASS-21 for the assessment of symptoms of depression, anxiety and stress allows greater reliability for the clinical use of this scale.
Reviewer 2 Report
9-16 remove
51 It might not be surprising to understand how anxiety and depression might be associated with direct superior commitment to ensure good working conditions. (ref: https://doi.org/10.3390/ijerph18189676 )
74 participants missing (inclusion criteria?). Design timeframe missing..
Follow CROSS checklist: https://www.researchgate.net/publication/353522424_Checklist_for_Reporting_Of_Survey_Studies_CROSS
204 Conventionally start the discussion with the paraphrase of the study objective
318 might indicate
Author Response
Dear Reviewer,
We would like to thank you all for your attention during the review process of our manuscript. We are very grateful for all the corrections and suggestions made, which contributed greatly to its improvement. We hope to have carried out all the adjustments requested by the reviewers and met their expectations. We remain at our disposal for any necessary clarification.
AUTHORS’ ANSWERS
We greatly appreciate your contributions. They are really contributing to improve our manuscript.
1. The sentences were removed from the abstract.
2. You are totally correct about that. Thank you for the consideration. This study was added in the Discussion.
5. Inclusion criteria and design timeframe were introduced. The CROSS checklist was analyzed.
4. Done.
5. Sorry, but we didn’t understand this suggestion.
Reviewer 3 Report
Thank you for the opportunity to review the paper entitled “Depression, Anxiety and Stress in Health Professionals in the COVID-19 Context”. Below are my comments:
Abstract
- Lines 9-16: Could the authors please check why these sentences “A single paragraph of about 200 words maximum. For research articles, abstracts should … exaggerate the main conclusions” were included in the abstract. Please remove those sentences.
- Line 24: The study participants were 529 health professionals (82.4% women and 66.7% Nursing professionals). This sentence is unclear – is it among 82.4%, 66.7% were nursing professionals? Why did the authors use capital letter for “Nursing”?
Introduction
- Lines 37-38: For the examples of the stressors faced daily, the authors should cite those examples that are related to COVID-19 so “low pay, lack of professional recognition” may not be appropriate.
- What is/are the gap(s) of the existing literature which the present study fills?
Results
- Why did the authors not compare the levels of depression, anxiety and stress among those with more years of working experience and those with/without children?
Discussion
- The discussion part is like a literature review on the prevalence estimates of depression and anxiety. How about other results such as lines 174-176?
Limitations
- Were the samples representative?
Minor corrections
- Line 84: … Health Council [20] – a full stop is missing.
- Line 93: Could you please replace “experience time”
- The authors are advised to consult professionals for language editing as some sentences were not complete sentences (e.g., line 197-199: Highlight the moderate to extreme severity of depression…among the health professionals.)
Author Response
Dear Reviewer,
We would like to thank you all for your attention during the review process of our manuscript. We are very grateful for all the corrections and suggestions made, which contributed greatly to its improvement. We hope to have carried out all the adjustments requested by the reviewers and met their expectations. We remain at our disposal for any necessary clarification.
AUTHORS’ ANSWERS
- Abstract
The lines 9-16 were removed from the abstract.
The capital letter for “Nursing” was removed.
- Introduction
Low pay and lack of professional recognition have been evidenced by the scientific literature as determinants of professional stress.
A scarcity of scientific evidence on the assessment of the prevalence of anxiety, depression and stress among health professionals has been identified. This is the relevance of our study.
- Results
We apologize, but this was not the focus of our study.
Round 2
Reviewer 3 Report
The authors need to address all the questions raised by the reviewer in the review report, e.g., 6, 7, 9 and 10. Thank you.
Author Response
Dear Reviewer,
We apologize for not having described these justifications in advance.
See below:
- Abstract
The lines 9-16 were removed from the abstract.
The capital letter for “Nursing” was removed.
- Introduction
Low pay and lack of professional recognition have been evidenced by the scientific literature as determinants of professional stress.
A scarcity of scientific evidence on the assessment of the prevalence of anxiety, depression and stress among health professionals has been identified. This is the relevance of our study.
- Results
We apologize, but this was not the focus of our study.
- Discussion
A scarcity of scientific evidence on the assessment of the prevalence of anxiety, depression and stress among health professionals using the DASS-21 was identified. If necessary, this information can be added to the discussion.
- Although not to large, the sample was representative of the population studied.
- We apologize for the mistakes initially made by the translators in the Portuguese-English version. In accordance with the requests of all reviewers, the manuscript was submitted to an extensive English revision.
I hope you understand our failure.
Kind regards,
The authors
